# Characterization of Rapeseed Oil Oleogels Produced by the Emulsion Template Method Using Hydroxypropyl Methylcellulose and the Drying Kinetics of the Emulsions

**DOI:** 10.3390/foods14162908

**Published:** 2025-08-21

**Authors:** Mario Lama, Amaya Franco-Uría, Ramón Moreira

**Affiliations:** Department of Chemical Engineering, Universidade de Santiago de Compostela, rúa Lope Gómez de Marzoa, s/n, 15782 Santiago de Compostela, Spain; mario.lama@usc.es (M.L.); amaya.franco@usc.es (A.F.-U.)

**Keywords:** binding capacity, canola oil, colour, drying modelling, oxidation, rheology, texture

## Abstract

Given health concerns, oleogels are promising substitutes for saturated fats in food products. An emulsion-templated method was used, employing rapeseed oil and hydroxypropyl methylcellulose (HPMC) as the structuring agent, to produce oleogels. Oil-in-water emulsions (50:50 *w*/*w*) were prepared with three HPMC concentrations (1.5, 2.0, and 2.5% *w*/*w*) and dried convectively at 60, 70, 80, and 90 °C to obtain oleogels. The emulsions exhibited viscoelastic behaviour with a predominant viscous character, G″ > G′. Drying kinetics showed a constant rate period followed by a falling rate period; the latter was satisfactorily modelled using a diffusion-based approach. All oleogels displayed predominantly elastic behaviour but the characteristics depended on the temperature employed during the drying operation and the HPMC content. The mechanical moduli (G″ and G′) of the oleogels increased significantly with a drying temperature below 80 °C. Higher HPMC content enhanced structural development and thermal stability. Most oleogels exhibited high oil binding capacity (>85%), which increased with the drying temperature and the HPMC content. A correlation was established between the elastic moduli, oil retention, and the hardness of the oleogels. No significant influences of the drying temperature and the polymer concentration on lipid oxidation and colour samples were determined. These results highlight the importance of selecting appropriate drying conditions based on the desired final product properties.

## 1. Introduction

Improving public health has long been a key goal of the food industry, where reducing solid fats in formulations is a priority [1]. While these fats provide desirable properties, such as texture, stability, and palatability, they are linked to adverse health effects, including notably increased LDL cholesterol and cardiovascular risk [2]. To address this, unsaturated fats from vegetable oils are being explored as alternatives, though they require structuring to replicate the functionality of saturated fats. This structuring leads to the formation of oleogels by incorporating gelling agents [3].

There are various methods for preparing oleogels. The direct method, involving the dissolution of gelators at high temperatures followed by cooling, is straightforward but unsuitable for food use due to oil oxidation [4]. Indirect approaches, especially the emulsion-templated method, are more prevalent. This procedure involves forming an oil-in-water emulsion stabilised by a structuring agent, then removing water through dehydration [5]. This step is critical, as both the drying technique and the operational conditions significantly affect the quality of the oleogel. Thus, the optimization of this operation is crucial for the economic viability of oleogel production. Several dehydration techniques have been reported in the literature. Moradabbasi et al. [6] compared vacuum oven-drying and freeze-drying, finding that freeze-drying produced more stable structures with less damage. Lama et al. [7] compared convective air-drying and freeze-drying, finding that the latter produced weaker structures with lower oil retention. Despite the fact that dehydration is mandatory in the template method, the impact of the drying conditions on the final oleogels properties is scarcely evaluated.

Regardless of the method, oleogel formation requires a structuring agent, often used in small amounts, with cellulose derivatives like hydroxypropyl methylcellulose (HPMC) widely employed [8]. This amphiphilic biopolymer, composed of methyl and hydroxypropyl groups, exhibits surface activity and distinct hydration/dehydration behaviour [9,10]. Promising results have been reported for the use of HPMC to produce oleogels. Saavedra et al. [11] obtained gels with high sunflower oil retention (>90%), which is a desirable outcome. However, a decrease in this parameter was observed with an increasing drying temperature, highlighting the importance of determining the adequate drying conditions. Furthermore, an increase in product hardness was reported with both high drying temperatures and structuring agent content, the latter being the more significant variable for this parameter. Additionally, Wei et al. [12] reported on the broad oxidative stability of perilla oil over time, which was attributed to its encapsulation within the HPMC matrix.

Rapeseed oil (also known as canola oil) stands out among vegetable oils for its low polymerization tendency and extended shelf life, exhibiting high oxidation stability [13]. While its structuring has primarily been studied using waxes in direct methods, few studies have addressed its use with HPMC. Wettlaufer et al. [14] analysed the differences between rapeseed oil structured with various waxes, finding that the presence of minor polar components in the oil promotes structuring, particularly when using waxes with low ester content (such as candelilla and sugarcane wax). Adrah et al. [15] used carnauba wax as a structuring agent and compared the frying performance between free and structured oil. They observed a significant reduction in fat content in the final product, although no improvement in oxidative stability was found after thermal treatment. Oh et al. [16] conducted a study using HPMC. The resulting solid was better structured than beef tallow (control) and had a lower fat content. Moreover, the entrapment of the oil within the oleogel matrix gave oxidative stability against heat, which improved with the increasing HPMC concentration.

Due to the advantages of rapeseed oil over other vegetable oils in terms of its properties, and the functional characteristics of HPMC, the present work aims to study the formation of rapeseed oil oleogels using HPMC as a structuring agent. This study focuses on the influence of the drying conditions (specifically, air-drying temperatures) and the HPMC content on the physical, rheological, textural, and oxidative properties.

## 2. Materials and Methods

### 2.1. Materials

For the preparation of the oleogels, hydroxypropyl methylcellulose (80–120 cP) in 2% aqueous solution at 20 °C was purchased from Sigma-Aldrich (St. Louis, MO, USA). Rapeseed oil (Aro, Düsseldorf, Germany) with a peroxide index <10 meq O_2_/kg and an acidity of 1% was employed. Chromatic coordinates of the rapeseed oil were *L** = 34.15 ± 0.52; *a** = −1.38 ± 0.06; *b** = 8.50 ± 0.11.

For the oxidation analysis, the following reagents were used: chloroform, potassium iodide, isooctane (VWR Chemicals, Leuven, Belgium), pentahydrate sodium thiosulfate (Panreac, Barcelona, Spain), and *p*-anisidine (Thermo Fisher Scientific, Bridgewater, NJ, USA).

### 2.2. Preparation of the Emulsions

The emulsions were prepared following the protocol described by [17] and slightly modified by [11]. Briefly, HPMC was dispersed in 50 g of rapeseed oil and stirred at 280 rpm using a paddle stirrer for 5 min. Subsequently, chilled water (~10 °C) was added over 30 s, followed by an additional 30 s to ensure proper dispersion. The emulsion was then homogenised in two continuous stages using a high-energy disperser (Ultra-Turrax T-25 Basic, IKA, Staufen, Germany): the first stage at 6500 rpm for 15 s, and the second at 17,500 rpm for 60 s. The initial oil-in-water emulsion composition consisted of 50% (*w*/*w*) rapeseed oil, HPMC at varying concentrations (1.5, 2.0, and 2.5% *w*/*w*), and chilled water added to reach 100%.

### 2.3. Drying of the Emulsion

After preparation, the emulsions were allowed to stand at room temperature (20 °C) for 20 min. Then, the emulsions were placed in Petri dishes (19.5 cm diameter), which had been previously weighed using a precision balance (Mettler PJ3000, Gemini BV, Apeldoorn, The Netherlands), until reaching a layer of 2.0 mm of thickness. The samples were introduced in a forced convective tray dryer (Angelantoni Challenge 250, Massa Martana, Italy) to remove the water until a dried, gelled solid was formed. The final moisture content of the system was less than 0.01 (kg water/kg dry solid, d.b.). Drying experiments were performed at four drying temperatures (60, 70, 80, and 90 °C) at constant air velocity and relative humidity (2 m/s and 10%, respectively).

The drying kinetics were obtained (*n* = 3) by weight monitoring of the samples. During the first hour, measurements were taken every 5 min and, as the drying progressed, and the rate decreased, they were subsequently taken every 20 min. The equilibrium moisture content was assumed to be zero in all the experiments due to the air conditions employed (high temperature and low relative humidity). The drying rates, expressed as −*dX/dt* (kg water/(kg dry solid·min)), were determined for each experimental condition and plotted as a function of the absolute moisture content *X* (kg water/kg dry solid). This approach provides a clearer visualization of the drying behaviour, particularly for distinguishing between the constant-rate and falling-rate periods. During the constant-rate stage, surface moisture is primarily removed. Under stable external air conditions, the drying rate in this phase can be described by Equation (1):(1)$$-\frac{dX}{dt}=k$$

The falling-rate period can be described using a diffusion model. This model is derived from Fick’s law, and the corresponding solution was established by [18]. This is valid for an infinite slab system and under the main assumptions of negligible external resistance, homogeneous initial conditions, and constant diffusivity. It describes the diffusion process of water during the drying stage, enabling the determination of the effective water diffusion coefficient through this modelling approach. Depending on the drying time, Equation (2) can be applied for long drying times, while Equation (3) is used for short drying times, as follows:(2)$$\frac{X_{c}-X}{X_{c}}=1-\sum_{n=0}^{\infty }\frac{8}{(2n+1{)}^{2}\pi^{2}}e^{{-D}_{eff}(2n+1{)}^{2}\frac{\pi^{2}t}{r^{2}}}\;\;\;\;\;\;\forall \left(\frac{X_{c}-X}{X_{c}}\right)>\;0.4$$(3)$$\frac{X_{c}-X}{X_{c}}=2{\left(\frac{D_{eff\;}t}{r^{2}}\right)}^{0.5}\left\{\pi^{-0.5}+2\sum_{n=1}^{\infty }(-1{)}^{n}ierfc\frac{nr}{\sqrt{D_{eff}\;t}}\right\}\;\;\;\;\;\;\forall \left(\frac{X_{c}-X}{X_{c}}\right)<0.4$$
where *D_eff_* represents the effective water diffusion coefficient (m^2^/s), *X_c_* is the critical moisture, and *r* is the thickness of the sample (m).

The goodness of fit between the experimental and predicted data was defined by R^2^ and the root mean squared error (RMSE).

### 2.4. Oleogel Preparation

After drying, the samples were left to stand for 30 min at room temperature. Subsequently, the dried solid was homogenised using the Ultra-Turrax disperser at 17,500 rpm for 90 s in 10-s pulses to obtain a homogeneous paste, placed in ice cube trays (5 g of oleogel were added to each compartment) and kept refrigerated at 4 °C for 12 h to obtain the oleogels. Subsequently, prior to characterization analyses, the samples were allowed to equilibrate for 20 min to ensure that the entire analysis is conducted at room temperature.

### 2.5. Rheological Characterization

The rheological characterization of the emulsions (*n* = 3) was performed using a stress-controlled rheometer (Anton Paar MCR301, Graz, Austria) with a plate-to-plate geometry (50 mm) and a 1.0 mm gap at a constant temperature of 25 °C. Initially, the linear viscoelasticity range (LVR) was determined by strain sweep from 0.01 to 10% at a constant frequency of 1 Hz. Subsequently, a frequency sweep from 0.1 to 10 Hz was performed with a 1% strain to determine the viscoelastic characteristics of the samples. Steady flow curves were performed varying from 0.001 to 1 s^−1^ of shear rate. Additionally, a heating ramp was performed from 25 to 90 °C at a constant strain of 1%, a frequency of 1 Hz, and a heating rate of 3 °C/min to determine the gelation temperature of the emulsions through complex viscosity analysis.

For the oleogels (*n* = 3), the same plate was used but with a 1.5 mm gap. The LVR was performed over the same strain range at a constant frequency of 1 Hz, while the frequency sweep used a 0.01% strain (the LVR of the oleogels was shorter than those measured in the emulsions) over the same range. Finally, a heating ramp was performed from 25 to 80 °C at 0.01% strain, a 1 Hz frequency, and a heating rate of 3 °C/min. In all experiments, the temperature was controlled by a Peltier system (±0.01 °C).

### 2.6. Textural Properties

Oleogel samples of 19 mm diameter and 8.0 mm height were subjected to a textural profile analysis test in a texturometer (TA.XT Plus, Stable Micro Systems, Surrey, UK) fitted with a cylindrical probe of 25 mm diameter (SMS P/25) and a 5 kg load cell. The samples were compressed to 50% of their original height. The initial applied force was 0.1 N, and the pre-test, test, and post-test speeds were set at 1.0, 2.0, and 1.0 mm/s, respectively. Four replicate samples were analysed for each batch. From this test, the following parameters were obtained: hardness, expressed as the maximum stress (kPa), was calculated by the maximum applied force on the sample (N) per initial compression area (m^2^) of the sample; adhesiveness (N·s), representing the force required to overcome the adhesive interaction between the sample and the probe; cohesiveness (dimensionless), indicating the material’s resistance to a second deformation; and elasticity (dimensionless), reflecting the sample’s ability to recover its original shape after deformation.

### 2.7. Oil Binding Capacity (OBC)

The oil binding capacity (*OBC*) of the oleogels was determined according to the method described by [19], with minor modifications as reported by [11]. Briefly, 1 g of oleogel was placed into pre-weighed Eppendorf tubes (*n* = 3). The samples were centrifuged using a minicentrifuge (HWLAB, HW12, Shiley, NW, USA) at 13,500 rpm for 25 min at 20 °C. After centrifugation, the supernatant oil was removed by inverting the tubes for 1 min, and the tubes were then weighed again. The *OBC* was calculated using Equation (4):(4)$$OBC\;\left(\%\right)=1-\left(\frac{m_{2}-m_{t}}{m_{1}-m_{t}}\right)\;\cdot 100$$
where *m_t_* (g) is the weight of the Eppendorf tube, *m*_1_ and *m*_2_ (g) are the weight of the Eppendorf with the sample pre- and post-centrifugation, respectively.

### 2.8. Oxidation Degree

#### 2.8.1. Primary Oxidation

Primary oxidation was assessed by determining the peroxide value of the sample, following [20]. Briefly, 1 g of oleogel was placed in a 250 mL flask, and 10 mL of chloroform was added to dissolve the fat content. Subsequently, 15 mL of glacial acetic acid and 1 mL of supersaturated potassium iodide solution were added (note: HPMC is not soluble in these reagents). The mixture was stirred for 1 min and then left to stand in the dark for 5 min without agitation. Afterwards, 75 mL of distilled water were added, and the sample was titrated with 0.01 N sodium thiosulfate using an automatic titrator (Hanna HI901 Color, Woonsocket, RI, USA). Once the titration was complete, the peroxide value, *PI*, was calculated in milliequivalents of active oxygen per kilogram of sample (meq O_2_/kg) using Equation (5):(5)$$PI\;\left(\frac{meq\;O_{2}}{kg}\right)=\;\frac{V\cdot C\cdot 1000}{m}$$
where *V* is the volume (mL) of sodium thiosulphate used in the titration, *C* is the concentration of sodium thiosulphate (0.01 N), and *m* is the mass of the sample (g). The factor 1000 is a conversion factor. This experiment was performed at least in duplicate.

#### 2.8.2. Secondary Oxidation

Secondary oxidation was determined by measuring the *p*-anisidine value according to [21] with slight modifications. Briefly, a *p*-anisidine solution (0.25% *w*/*v*) was prepared in glacial acetic acid. A total of 1 g of oleogel was placed in a 10 mL volumetric flask and brought to volume with isooctane. The mixture was then subjected to an ultrasonic bath (Elmasonic S40H, Singen, Germany) for 5 min to dissolve the sample. Subsequently, 1 mL of the resulting solution was transferred to a test tube, followed by the addition of 0.2 mL of the *p*-anisidine solution. The mixture was vortexed to homogenise the solution and left to react in the dark for 10 min. After the reaction time, the absorbance of both the original (blank) and the reacted solutions was measured at 350 nm using a UV-VIS spectrophotometer (Thermo Fisher Spectronic Genesys 10 UV, Waltham, MA, USA). The *p*-anisidine value (*AV*) was then calculated according to Equation (6):(6)$$AV=\frac{10\cdot (1.2As-Ab)}{m}$$
where *As* is the absorbance of the reacted solution, and *Ab* is the absorbance of the blank. This experiment was performed at least in duplicate.

### 2.9. Colour

The colour of the oleogels (*n* = 9) was measured using a colorimeter (Konica Minolta CR-400, Osaka, Japan), which was previously calibrated with a white reference plate. The colour assessment was provided in terms of CIELAB colour space coordinates which correspond to lightness (*L**), red–green (*a**), and yellow–blue (*b**). To evaluate the total colour difference, Δ*E*, Equation (7) was employed as follows:(7)$$∆E=\;\sqrt{{\left(L^{*}-L_{0}^{*}\right)}^{2}+{\left(a^{*}-a_{0}^{*}\right)}^{2}+{\left(b^{*}-b_{0}^{*}\right)}^{2}}$$
where *L**, *a**, and *b** are the colour coordinates of the sample and subscript 0 corresponds to the colour of a reference sample (oleogel with 1.5% (*w*/*w*) of HPMC dried at 60 °C).

### 2.10. Statistical Analysis

The statistical analysis was performed using IBM SPSS Statistics 29 software (IBM Corp., Armonk, NY, USA). A general linear model (GLM) approach was applied to assess the variance at a confidence level exceeding 95% across all dependent variables evaluated in this study, including textural parameters, oil binding capacity (*OBC*), oxidation, and colour. The model considered the drying temperature and the HPMC concentration as independent variables.

## 3. Results and Discussion

### 3.1. Characterization of the Emulsion

The same rheological features of the emulsions at different HPMC concentrations were obtained. In Figure 1a, corresponding to the strain sweep—used to evaluate the linear viscoelastic region (LVR)—it is observed that G″ remained slightly higher than G′ throughout the entire strain range, indicating that the emulsions exhibit a predominantly viscous character, and suggesting that the emulsion stability is low [22]. Additionally, both moduli tended to increase as the concentration of the gelling agent increased, requiring greater force to deform the emulsion. A similar trend was reported by Alizadeh et al. [23], who observed that increasing the concentration of the gelling agent promotes the formation of a multilayer biopolymer network that entraps oil droplets, resulting in a more structured emulsion. Moreover, the LVR is broad at the three HPMC content, exceeding 1% strain and wider with increasing gelling agent content. For example, in the emulsion containing 1.5% *w*/*w* HPMC, the LVR extended up to 1.5% strain, whereas at 2.5% *w*/*w*, it was up to 4.6%. This result indicated greater resistance to deformation, attributable to improved structural organization. This enhanced structuring was further confirmed by the frequency sweep tests (Figure 1b), which revealed trends like those observed in the strain sweep. Across the entire frequency range, G″ remained slightly higher than G′, again indicating restricted emulsion stability. Both moduli showed a slight increase with the increasing gelling agent concentration. According to Tadros [24], emulsions with a dispersed phase volume fraction below 0.56 are expected to exhibit more viscous than elastic behaviour, which is consistent with the tested emulsions, with oil volume fractions ranging from 0.515 to 0.525 as the HPMC concentration increased. Notably, the most concentrated emulsion, which had the highest volume fraction, also exhibited the smallest differences between G′ and G″, indicating greater structural integrity compared to the others.

Similarly, in the flow curve (Figure 1c), a clear increase in the apparent viscosity was observed with the increasing gelling agent concentration. The emulsions exhibited pseudoplastic behaviour, as evidenced by the decrease of the apparent viscosity with the increasing shear rate. Nevertheless, Newtonian plateaus were observed at both low and high shear rates, being more pronounced at low shear rates. Espert et al. [25], who conducted a comparative study of different O/W emulsions (50/50 wt) using hydroxypropyl methylcellulose (HPMC) and methylcellulose (MC), reported similar behaviour. It was noted that both cellulose ethers tend to exhibit Newtonian behaviour at low shear rates and, as the shear rate increases, their viscosity decreases, indicating pseudoplastic behaviour. The viscosity range obtained for the HPMC emulsion was between 1300 and 150 Pa·s (within a shear rate range of 0.001 to 1 s^−1^), which is higher than the values observed in this work (ranging from 333 to 20 Pa·s for the most concentrated and viscous system, considering the same shear rate range). This difference is mainly attributed to the higher viscosity grade of the HPMC used by [25] (4000 cP), compared to the grade employed in this research (80–120 cP). Finally, the emulsions exhibited hysteresis loops between upwards and downwards flow curves, indicating a certain degree of structural breakdown (viscosity loss) and a restructuring time required for the emulsions to recover their internal structure. This hysteresis decreased with the increasing HPMC concentration, suggesting that, in more concentrated systems, molecular interactions reestablish their structure more rapidly and efficiently after shear stress is applied.

Figure 1d shows the thermal trend of the complex viscosity of the emulsion with 2.5% (*w*/*w*) of HPMC, as an example. Initially, as heating begins, the complex viscosity is slightly decreased, likely due to partial structural disorganization, a phenomenon commonly observed in HPMC-based systems under thermal stress [9]. Nevertheless, this trend stabilises around 50 °C, followed by a progressive increase in viscosity up to approximately 70 °C. This increase in complex viscosity is directly associated with the gelation process of the system. According to Montes et al. [26], aqueous HPMC systems undergo gelation within a temperature range of approximately 60–75 °C. which depends on the propyl/methyl substitution ratio in the HPMC structure (the same HPMC is used in both studies). When forming an oil-based system with HPMC, a similar gelation temperature range is observed, slightly reduced to between 50 and 70 °C. Beyond this temperature, the slope of the curve decreased, indicating that a well-structured gel was already formed. Consequently, it can be inferred that oleogel formation within this gelation range temperature may be more complex than at higher temperatures.

### 3.2. Drying Kinetics

Figure 2 presents the drying kinetics of the emulsions with the different HPMC content at different drying temperatures. In all cases, the moisture content initially decreased rapidly, followed by a progressive slowdown as the drying process continues. This behaviour is attributed to the initial evaporation of free water, which is readily removed from the emulsion, and the subsequent evaporation of bound water, which is more tightly retained within the hydrocolloid matrix [11]. This trend is consistent with typical drying curves, where an initial constant-rate period is observed until reaching the critical moisture content, after which the drying rate declines and the process enters the falling-rate period.

As expected, higher drying temperatures led to shorter overall drying times, a trend reported in various matrices, including oleogels [7,11], food systems [27,28], and biomass [29]. For example, in the least concentrated system (1.5% HPMC), the drying times at 70, 80, and 90 °C were 180, 97, and 83 min, respectively. However, a clear difference was observed between the drying kinetics at 70 °C and those at higher temperatures. Two main factors may explain this behaviour. Firstly, the drying temperature directly influences water removal rates. Saavedra et al. [11] reported that the evaporation rate of free water increases significantly with temperature (0.033 kg water/(kg dry solid·min) at 70 °C, 0.041 kg water/(kg dry solid·min) at 80 °C, and 0.045 kg water/(kg dry solid·min) at 90 °C). Secondly, gelation of the HPMC phase plays a crucial role. As observed in the temperature ramp test of the emulsion (Figure 1d), gelation occurs between 50 and 70 °C. Thus, drying at a temperature within this range may obstruct water release due to the structural transitions associated with gel formation.

By contrast, at 80 °C and 90 °C, drying is performed above the gelation temperature. In this state, a syneresis phenomenon may occur, defined as the expulsion of water from a gel matrix during or after gelation [30]. This release of free water, combined with high drying temperatures and elevated evaporation rates, accelerated significantly the drying process. However, the kinetics at 60 °C were not satisfactorily determined due to a lack of experimental reproducibility. This could be attributed to the use of a drying temperature close to the gelation onset temperature of HPMC, as shown in Figure 1d.

On the other hand, the critical moisture content exhibited slight variations across the different concentrations and drying temperatures, remaining around 0.4 kg water/kg dry solid.

Regarding the HPMC concentration, slower drying was observed as the HPMC concentration increased. This trend was expected, as higher polymer content leads to a more structured system, which was confirmed by a rheological analysis of the fresh emulsions. The formation of more compact structures prevents the movement of water through them. This observation is supported by previous studies employing different structuring agents, such as HPMC [11] and chitosan [7].

### 3.3. Drying Kinetics Modelling

Drying kinetics modelling is essential for predicting drying times under different conditions and industrial dryer design. The constant-rate period was modelled using Equation (1). Table 1 presents the experimental rate constants, *k* (kg water/(kg dry solid·min), for the emulsions at the studied HPMC concentrations and drying temperatures. A significant decrease was observed with the increasing HPMC concentration (*p* < 0.05) from 1.5 to 2.0 (%, *w*/*w*). This behaviour is expected, as previously discussed in the kinetics description, since a higher HPMC concentration leads to a more structured matrix, which retains water more effectively. Regarding temperature, a significant increase in *k* was observed (*p* < 0.05), mainly between 70 and 80 °C. This can be attributed to the gelation phenomenon occurring during the process, as previously explained. Furthermore, the increase in the drying rate during the constant-rate period from 80 to 90 °C was attenuated by the HPMC concentration, with the most pronounced difference observed in the least concentrated emulsion, and no significant differences (*p* > 0.05) were observed between emulsions with an HPMC concentration above 2.0% (*w*/*w*) dried at 90 °C. Saavedra et al. [11], using HPMC concentrations ranging from 1 to 3% (*w*/*w*) and dry temperatures between 70 and 100 °C, reported similar values for constant drying rates.

The falling-rate period was modelled using a diffusional approach considering that the dominant mechanism for water removal is water diffusion through the matrix formed by the structuring agent. For this purpose, Equations (2) and (3) are applied to both the short- and long-time analytical solutions, respectively. The characteristic length used for modelling was the thickness of the sample, which was considered constant for each drying kinetic. The fitting was performed using the Solver tool in MS Excel, applying the least squares method. This allowed the determination of the effective water diffusivities for each HPMC concentration and drying temperature, as shown in Table 1. The drying curves predicted by the diffusional model (below critical moisture content, 0.4 kg water/kg dry solid) are plotted in Figure 2. The goodness of fit of the diffusional model was satisfactory, as evidenced by the statistical parameters R^2^ (>0.98) and RMSE (<0.087), except for 1.5% (*w*/*w*) of the HPMC content and the temperature of 70 °C where the statistical parameters were worse, Table 1.

The water diffusivity values showed the expected trend with temperature, with a significant increase at a constant HPMC concentration (*p* < 0.05), mainly between 70 and 80 °C. This trend aligns with the behaviour observed during the constant-rate period and the corresponding *k* values. The range of water diffusivity values agreed with those reported in other studies involving oleogels [7] and other solids, such as fruit [31,32].

Regarding the HPMC content, no clear trend was observed across the entire range studied. When the concentration increased from 1.5 to 2.0% (*w*/*w*), a significant decrease in the effective diffusivity of water was observed. However, when the concentration increased from 2.0 to 2.5% (*w*/*w*), the previous trend with HPMC content remained only at 70 °C. At 80 and 90 °C, no significant differences (*p* > 0.05) were observed between the corresponding water diffusivity values. This result seems to indicate that the resistance to water movement through the gelling structure is similar in this range of drying temperatures regardless of the HPMC concentration.

### 3.4. Characterization of Oleogels

#### 3.4.1. Rheology

In Figure 3, the strain and frequency sweeps of the oleogels dried at 60 and 80 °C (the other temperatures exhibited the same trend) are shown for all tested HPMC concentrations. From the strain sweep, the oleogels exhibited a predominantly elastic behaviour (G′ > G″) across the entire strain range, which is typical of structured gels. Both moduli increased with the HPMC concentration due to the improved structuring of the three-dimensional network that entraps the oil [33,34]. On the other hand, the LVR ranges were shorter than those determined in the emulsions, reaching approximately 0.1% strain. This is attributed to the higher elastic character of the oleogels (at a higher HPMC content, a shorter LVR, for example, in samples dried at 60 °C, the range was shortened from 0.15 up to 0.08, increasing the HPMC content from 1.5 to 2.5% *w*/*w*), which advances the region where irreversible deformation takes place. On the other hand, both moduli values tended to increase as the drying temperature increased from 60 to 80 °C at the same HPMC concentration and subsequently decreased at 90 °C.

In the frequency sweep, it is again observed that G′ > G″ (by approximately an order of magnitude) across the studied frequency range. Furthermore, as indicated in the strain sweep, moduli values increased with the HPMC concentration, which is expected, and indicates stronger gels. Table 2 displays the elastic modulus values at 1 Hz for the different systems. In all cases, G′ increased significantly (*p* < 0.05) with the HPMC concentration. The drying temperature employed to form the oleogels also affected the viscoelastic behaviour of the gel. In the interval from 60 to 80 °C, more rigid gels were obtained, but at 90 °C, the elastic modulus dropped significantly (with values near to those obtained after drying at 60 °C), resulting in weaker gels. A possible explanation of this unexpected result could be related with the syneresis phenomena during gelation, promoted by the temperature, which decreased the available water to form the structured network giving, as a result, a structural weakening of the gel.

The low slopes observed in the frequency response indicated a weak frequency dependence, which implies good structural organization and the presence of strong gel networks [33,35]. However, as the HPMC concentration decreased (<2.5% (*w*/*w*)), G″ showed a slight increase in slope at higher frequencies. Consequently, in these oleogels, the damping factor (G″/G′) increased from 0.12 up to 0.18 with increasing frequency, while at the highest HPMC content, the damping factor was always less than 0.10.

The thermal sweep from 25 to 80 °C is shown in Figure 4. The predominantly solid-like behaviour remained during the heating ramp independently of the HPMC concentration and drying temperature. In fact, the viscoelastic moduli were practically unchanged with temperature (without thermal transitions), indicating that the studied systems are thermostable, regardless of the HPMC concentration or the drying temperature [36]. Wang et al. [33] reported high thermal stability in both HPMC and MC oleogels, in agreement with the results found here.

#### 3.4.2. Texture

Texture parameters provide important information regarding the physical characteristics of the product and are relevant for the potential applications of the oleogel. Figure 5 shows the results for hardness, adhesiveness, cohesiveness, and elasticity of the tested oleogels. At a constant HPMC concentration, hardness depended on the drying temperature and ranged from 3.35 to 6.21 kPa for the lowest HPMC concentration oleogel, and from 12.09 to 16.66 kPa for the most concentrated system. In this way, hardness increased significantly (*p* < 0.05) with the increasing HPMC concentration. This trend was expected, as a higher concentration of the oleogelator strengthened the three-dimensional network, resulting in harder structures [37]. These findings were consistent with previous studies on oleogels. Espert et al. [17] performed penetration tests and reported a significant increase in hardness when the HPMC concentration was raised from 1 to 2%, with values ranging from 20.63 to 43.93 kPa. Similarly, Farooq et al. [38] observed a notable increase in hardness with the structuring agent concentration. Regarding the drying temperature employed for the oleogel production, it was observed that hardness increased with temperature, showing a significant rise between 60 and 70 °C, with a less marked increase at 80 °C. This trend agreed with previous findings in other HPMC oleogels [11]. However, at 90 °C, a decrease in hardness was noted. This may be due to partial structural damage related to high temperature, which reduces the mechanical resistance to compressive work during TPA testing [7].

Adhesiveness ranged from 0.27 to 0.87 N∙s and exhibited an opposite trend to hardness, decreasing in general with both the temperature and the HPMC concentration. As the firmness of the samples decreased, a small amount of oil tended to be released towards the sample surface resulting in greater adhesiveness. The findings in the literature on this property are variable, with some studies reporting similar trends between hardness and adhesiveness [39].

Cohesiveness varied in a narrow range (from 0.24 to 0.42), and showed an inverse trend relative to hardness, with significant differences between the lowest concentration and the others (*p* = 0.028). By contrast, the drying temperature had a slightly significant effect, showing significant differences only at the highest HPMC concentration (*p* = 0.02). This behaviour can be explained considering that cohesiveness is the relationship between the forces required for the first and second compressions. As the samples become harder, the force required for compression increases, resulting in lower cohesiveness values. Different trends for this parameter, as a function of the structuring agent employed, have also been reported in the literature [40,41].

Finally, elasticity (ranged from 0.38 up to 0.76) did not show a clear trend (given by a relatively high dispersion of data, exhibiting a coefficient of variation (%CV) between 0.94% and 35%), without significant differences (*p* > 0.05) related to concentration or the drying temperature effects. In the literature, this parameter does not exhibit a consistent trend. For instance, Farooq et al. [38], working with chitosan, reported a similar range of elasticity (0.38 to 0.62), with a tendency to increase with the gelling agent concentration. However, Lama et al. [7], also using chitosan, observed the opposite trend, with elasticity decreasing as the concentration increased. Moreover, the effect of temperature was not clearly defined. On the other hand, Armijo et al. [41], who used ethylcellulose with the direct method, found a decreasing trend in elasticity with increasing EC concentration.

#### 3.4.3. Oil Binding Capacity

The oil binding capacity (*OBC*) is a key parameter in oleogels, as their primary functions are to structure and preserve the oil. Therefore, achieving the highest possible oil retention at room temperature is a central objective. Table 2 shows the *OBC* results for the tested oleogels dried at different drying temperatures. Firstly, the higher HPMC concentrations led to increased *OBC* values. This trend was expected, considering the described rheological and textural properties previously discussed, since higher concentrations of structuring agent enhanced the structure of the gelling matrix in which the oil is entrapped. Similar findings have been reported in the literature. For instance, Espert et al. [17] observed increased oil retention with rising concentrations of cellulose-derived oleogelators (HPMC and MC). Likewise, Zhu et al. [42] formulated oleogels using xanthan gum, achieving the highest oil retention (99%) in the most concentrated system (2% structuring agent). In another study, by Farooq et al. [38], oleogels made from chitosan and vanillin exhibited *OBC* values close to 95% at the highest chitosan concentration (2%). Regarding the influence of the drying temperature of the oleogels on the *OBC*, an improvement is generally observed with increasing the temperature from 60 to 80 °C. Nevertheless, at 90 °C, the *OBC* dramatically decreased. Therefore, again, a break in the trend of the thermal behaviour of the oleogel properties has been found at the highest drying temperature tested, and a previous hypothesis to explain this fact can also be employed here. In this case, other authors have also found this trend [11], who reported a reduction in oil retention in HPMC–sunflower oil systems when the drying temperature increased from 80 to 100 °C. Overall, most oleogels studied (with the exception of some samples with a low HPMC content) showed good oil retention, with most values exceeding 85%.

Moreover, this parameter could be successfully correlated with other properties, such as the elastic modulus and hardness. Figure 6 shows the trend of *OBC* and hardness with respect to the elastic modulus at 1 Hz. In general, clear relationships among the three variables can be observed, meaning that oleogels with a high elastic modulus showed high hardness and *OBC* values. This result is consistent with the underlying mechanisms previously discussed for each parameter.

#### 3.4.4. Oil Oxidation

The measurement of the oxidation level of oil in the final oleogels is of great importance, as it provides valuable insights into the quality of the final product. Oxidation has a significant impact on the sensory, nutritional, and shelf-life properties of foods, as well as on their overall safety for consumption [43]. In fact, the intake of products with high oxidation levels may pose serious risks to human health [44]. Figure 7 shows the results of both primary and secondary oxidation. Overall, all oleogels exhibited acceptable oxidation values, considering that the threshold set by the FAO for refined vegetable oils is 10 meq O_2_/kg [45]. In fact, the peroxide index (*PI*) ranged from 4.7 to 7.7 meq O_2_/kg, which was slightly higher than the value determined in the fresh rapeseed oil used for oleogel manufacture (4.5 ± 0.3 meq O_2_/kg). Similarly, the *p*-anisidine values varied from 2.7 to 5.5, also slightly exceeding the value of the fresh oil (2.5 ± 0.3). These results suggested that, during the production of oleogels, the oil had a relevant degree of protection by the HPMC structuration, particularly during the drying stage, in which high employed temperatures could promote extensive oxidative degradation. However, no significant differences (*p* > 0.05) were found in the results of the primary oxidation, and no clear or consistent trend was observed in the secondary oxidation with respect to the HPMC concentration. Initially, it was hypothesised that a high concentration of gelator would lead to increased oxidation, as the drying process becomes slower and the material is exposed to thermal treatment for long periods of time. This result has been reported by previous studies. For example, Miao et al. [46] studied the influence of water content in emulsions subjected to hot air and freeze-drying and found lower PI values when a lower water content was used, due to a faster drying process under both methods. Similarly, Lama et al. [7] reported higher peroxide values in oleogels with higher oleogelator concentrations. With respect to the drying temperature, no clear trend was observed in any of the oxidation measurements. However, in this case, an antagonistic interaction emerges between the drying temperature and time. When drying occurs at higher temperatures, the duration of the process is usually shorter, and oxidation depends on both variables. Individually, a long drying time and high temperatures would tend to increase the oxidation levels. However, in drying processes where the objective is to remove moisture without setting a fixed duration, the effect of temperature alone in the oil oxidation did not follow a consistent pattern, and considering the oxidation levels (and narrow range) achieved in the oleogels, the selection of the drying temperature to produce them can be considered not to be a relevant factor.

Finally, a comparison between direct and indirect methodologies highlights the importance of method selection. Armijo et al. [41] conducted a study on oleogels using the direct method and ethylcellulose as the structuring agent. They reported low peroxide values (ranging from 1.6 to 15.8 meq O_2_/kg); however, the *p*-anisidine values were significantly higher (between 10 and 70). This can be attributed to the fact that, as oxidation progresses, peroxides decompose into secondary oxidation products. Therefore, in oxidised samples, peroxide values tend to decrease while the *p*-anisidine values increase—an effect observed in the direct method. These results suggest that the indirect methodology offers better oxidative stability for oleogels during production

#### 3.4.5. Oleogels Colour

The colour measurements are presented in Table 2. Colour coordinates varied in a very narrow range (*L** from 28.8 ± 1.1 to 30.0 ± 2.7, *a** from −2.5 ± 0.1 to −2.2 ± 0.1, and *b** from 10.3 ± 0.4 to 11.9 ± 0.5). Assuming the colour coordinates of oleogel with 1.5% (*w*/*w*) dried at 60 °C as a reference, the maximum total colour difference determined was 1.5, that is less than 3. This last value is considered the minimum value to observe colour differences by the human eye [47]. Consequently, it can be concluded that the HPMC content and the drying conditions are irrelevant on the final oleogels colour features.

Saavedra et al. [11] reported *L** values ranging from 28 to 35 for a system containing 2% (*w*/*w*) HPMC. When the concentration increased (3% *w*/*w*), the *L** values decreased to a range between 23 and 30, although they consistently increased with rising temperature. The *a** coordinate decreased with the increasing HPMC concentration (from a range of −0.23 to −1.45 down to −0.9 to −1.2), while *b** increased (from 0.5–2 up to 2.2–3). This indicates that thermal processing causes a slight shift toward more greenish and yellowish tones, suggesting that oil oxidation due to heat may be responsible for the observed colour changes. However, it was concluded that colour variation was not particularly relevant. When compared with the results obtained in the present study, the *L** and *a** values are similar, but the *b** coordinate is considerably higher in oleogels made with rapeseed oil. This difference is attributed solely to the initial oil colour, as the *b** value of rapeseed oil is significantly higher than that reported for sunflower oil (8.50 vs. 1.84, respectively).

Moradabbasi et al. [6] compared the colour changes resulting from different drying techniques (freeze-drying vs. convective oven-drying) and found that oven-drying led to greater and differently directed colour changes: while freeze-dried samples showed decreasing *a** and *b** values, these parameters increased under convective oven-drying. This was associated with the thermal effect of the convective oven.

Furthermore, it is noteworthy that no significant darkening occurred in the samples after drying, contrary to the findings in other studies. Lama et al. [7] observed marked darkening due primarily to the oxidation of vanillin present in the system. By contrast, in the current case, the HPMC did not undergo substantial colour changes, resulting in only slight variations in the *L** coordinate.

## 4. Conclusions

This study established the link between rapeseed oil oleogel quality and two main independent variables, the HPMC content and the drying temperature. The rheological analysis of the emulsions indicated that the system becomes more structured with the increasing HPMC content. Both the drying temperature and the HPMC content influenced the drying kinetics, structuring stability, textural parameters (except elasticity) and the *OBC*. No influence of both variables was observed on the oleogel colour and oxidation values. In general, the increasing HPMC content resulted in higher *OBC* values and harder gels, while an opposite trend was obtained for cohesiveness and adhesiveness. Oleogel properties dried at 60 and 80 °C showed the same trend, although at 90 °C the characteristics were like those samples dried at low temperature (60 °C). The drying kinetics showed an initial constant drying rate period (critical moisture content around 0.4 kg water/kg dry solid) followed by a falling drying rate period. Shorter drying times were obtained when a low HPMC content was employed. Regarding the drying temperature, the HPMC gelation significantly influenced the drying kinetics, causing instability at 60 °C, and no significant differences were observed above 80 °C with the HPMC content higher than 2.0% *w*/*w*. The falling-rate period was successfully modelled employing a simplified diffusion-based model. Effective water diffusivities ranged from 3.0 × 10^−11^ m^2^/s (at 70 °C and 2.0% *w*/*w* HPMC) to 1.3 × 10^−10^ m^2^/s (at 90 °C and 1.5% *w*/*w* HPMC). In short, the drying temperature of 80 °C is recommendable because it prevents the gelation issues observed at 60–70 °C, and higher temperatures (90 °C) do not improve the oleogel properties. Concerning the HPMC content, the optimal concentration depends on the final function of the product: a high HPMC content (>2.0% *w*/*w*) is recommended to obtain a firm gel with high oil retention, whereas a low concentration is more suitable for a spreadable product. Finally, for future work, it would be of interest to conduct a stability study under different storage conditions, with the aim of evaluating the evolution of product properties and determining the optimal storage conditions. In this context, an oxidation analysis is essential to evaluate the shelf life of the samples.

## Figures and Tables

**Figure 1 foods-14-02908-f001:**
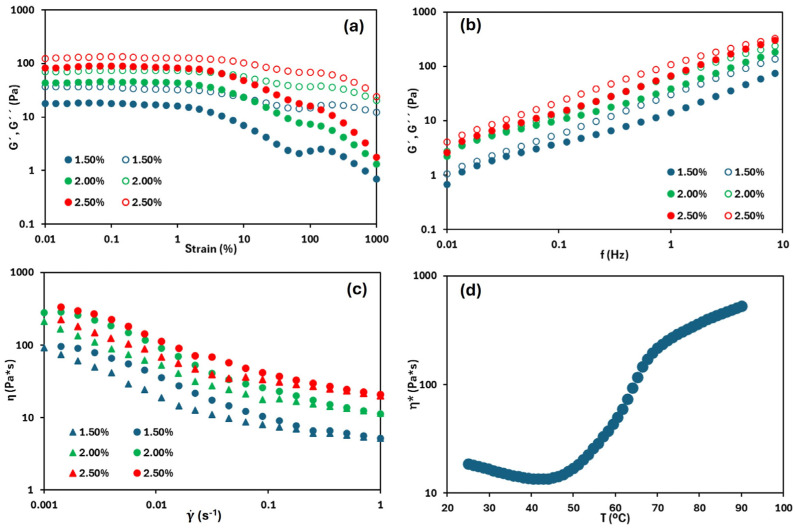
(**a**) Strain sweeps, (**b**) frequency sweeps. Filled symbols represent G′, open symbols represent G″. (**c**) Flow curve (circles indicate upwards curve, and triangles downward curve). (**d**) Complex viscosity versus temperature of the emulsion with 2.5% (*w*/*w*) HPMC.

**Figure 2 foods-14-02908-f002:**
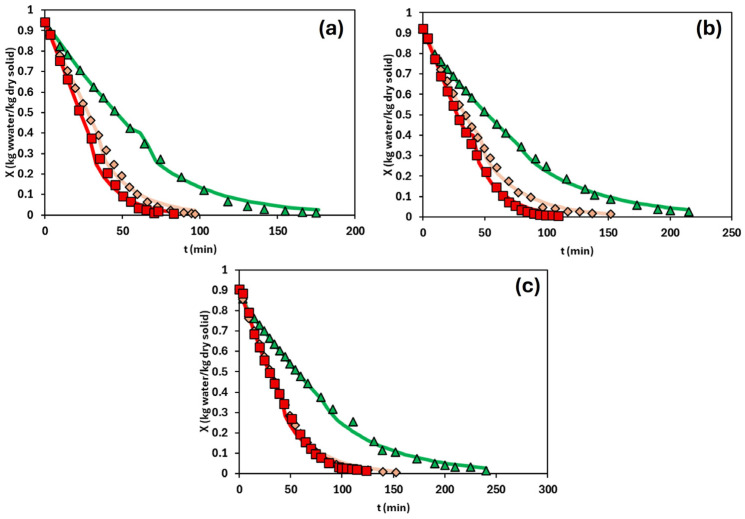
Drying kinetics of the emulsions at several drying temperatures. Plots correspond to (**a**) 1.5% (*w*/*w*), (**b**) 2.0% (*w*/*w*), and (**c**) 2.5% (*w*/*w*) HPMC content. Green triangles represent 70 °C, orange diamonds 80 °C, and red squares 90 °C. Lines represent the model’s prediction.

**Figure 3 foods-14-02908-f003:**
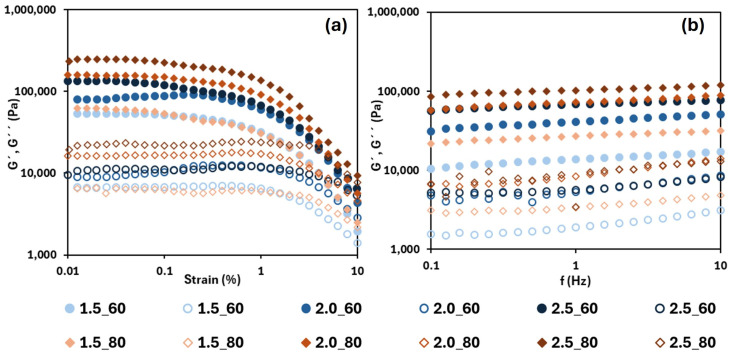
Strain sweeps (**a**) and frequency sweeps (**b**) for oleogels tested at 60 °C (circles) and 80 °C (diamonds). G′ is shown with filled symbols, and G″ with open symbols. 1.5, 2.0 and 2.5 are the HPMC concentration (%, *w*/*w*).

**Figure 4 foods-14-02908-f004:**
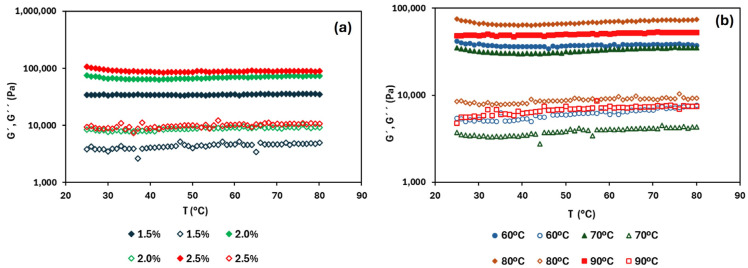
Heating ramp of tested oleogels: (**a**) different HPMC concentrations at a drying temperature of 80 °C; (**b**) different drying temperatures (60, 70, 80, and 90 °C) at a fixed HPMC concentration of 2.0% (*w*/*w*). Filled symbols represent G′ and open symbols represent G″.

**Figure 5 foods-14-02908-f005:**
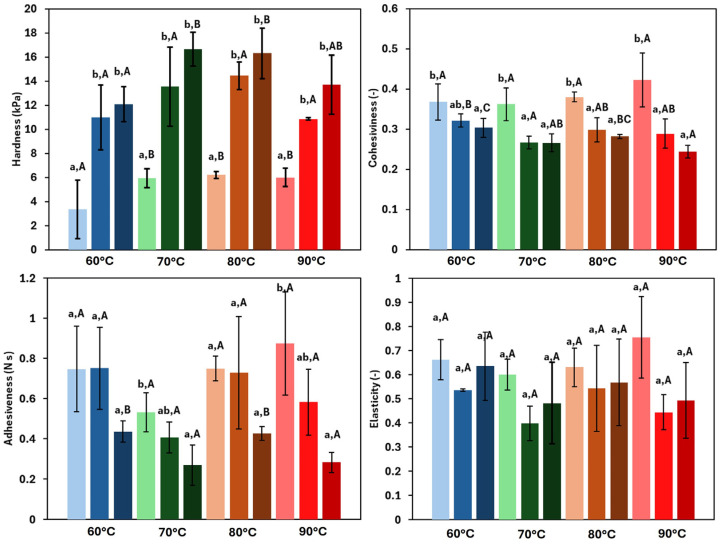
Textural properties of the oleogels. Samples dried at 60 °C (blue), 70 °C (green), 80 °C (orange), and 90 °C (red). Light colours represent 1.5% HPMC concentration, medium colours 2.0% HPMC concentration, and dark colours 2.5% HPMC concentration. Lowercase letters (a, b) indicate statistically significant differences (*p* < 0.05) between gelator concentrations at each drying temperature. Capital letters (A, B, C) indicate statistically significant differences (*p* < 0.05) between the drying temperatures at each gelator concentration. In both cases, based on Duncan’s test, with comparisons made independently, performed for each variable.

**Figure 6 foods-14-02908-f006:**
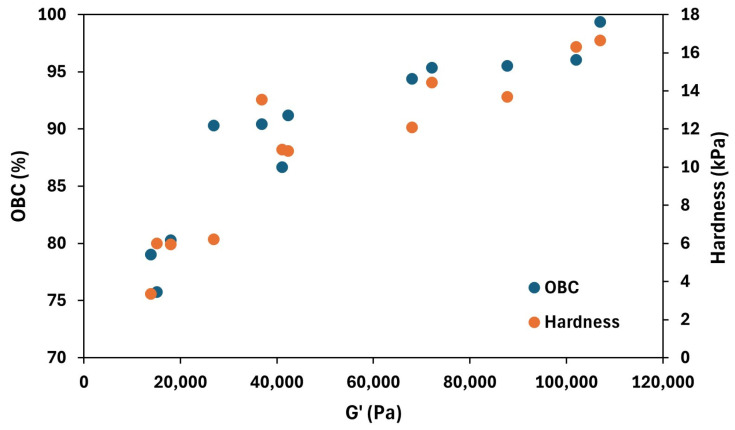
Relationship between the elastic modulus (G′), the oil binding capacity (*OBC*), and hardness of the tested oleogels.

**Figure 7 foods-14-02908-f007:**
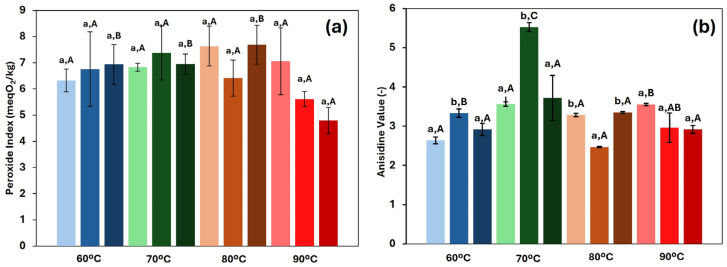
Primary (**a**) and secondary (**b**) oxidation for each of the systems studied at different drying temperatures. Light colours represent 1.5, medium colours 2.0, and dark colours 2.5 (%, *w*/*w*) HPMC concentration. Lowercase letters (a, b) indicate statistically significant differences (*p* < 0.05) between gelator concentrations at each drying temperature. Capital letters (A, B, C) indicate statistically significant differences (*p* < 0.05) between the drying temperatures at each gelator concentration. In both cases, based on Duncan’s test, with comparisons made independently, performed for each variable.

**Table 1 foods-14-02908-t001:** Constant drying rate for the different studied systems ^1,2^.

HPMC Content (%, *w*/*w*)	Temperature (°C)	*k* (10^−3^ kg Water/(kg Dry Solid·min))	*D_eff_* (10^−11^ m^2^/s)	R^2^	RMSE
1.5	70	8.9 ± 0.2 ^c,A^	5.21 ^c,A^	0.890	0.724
	80	15.9 ± 0.2 ^c,B^	9.81 ^b,B^	0.993	0.073
	90	19.3 ± 0.5 ^b,C^	12.8 ^c,C^	0.994	0.087
2.0	70	7.0 ± 0.4 ^b,A^	3.04 ^a,A^	0.994	0.040
	80	11.4 ± 0.2 ^a,B^	6.85 ^a,B^	0.989	0.045
	90	12.7 ± 0.3 ^a,C^	10.3 ^b,C^	0.982	0.054
2.5	70	6.24 ± 0.3 ^a,A^	3.63 ^b,A^	0.982	0.052
	80	12.6 ± 0.3 ^b,B^	6.89 ^a,B^	0.992	0.053
	90	12.7 ± 0.4 ^a,B^	7.69 ^a,C^	0.989	0.053

^1^ Lowercase letters (a, b, c) indicate statistically significant differences (*p* < 0.05) between gelator concentrations at each drying temperature. Capital letters (A, B, C) indicate statistically significant differences (*p* < 0.05) between the drying temperatures at each gelator concentration. In both cases based on Duncan’s test, with comparisons made independently, performed for each variable. ^2^ Standard deviations of *D_eff_* values are ±0.02.

**Table 2 foods-14-02908-t002:** Elastic modulus (G′) at 1 Hz, oil binding capacity (*OBC*), colour coordinates (*L**, *a**, *b**), and Δ*E* values of tested oleogels ^1^.

HPMC Content (%, *w*/*w*)	Drying Temperature (°C)	G’ (10^4^ Pa)	*OBC* (%)	*L**	*a**	*b**	Δ*E*
1.5	60	1.39 ± 0.04 ^a,A^	79.0 ± 1.7 ^a,AB^	30.09 ± 2.07 ^a,A^	−2.38 ± 0.25 ^a,A^	10.88 ± 0.79 ^a,AB^	0
	70	1.80 ± 0.05 ^a,C^	80.3 ± 2.6 ^a,B^	31.50 ± 2.70 ^b,A^	−2.19 ± 0.13 ^b,B^	10.33 ± 0.38 ^a,A^	1.53
	80	2.69 ± 0.07 ^a,D^	90.3 ± 1.4 ^a,C^	30.31 ± 0.61 ^b,A^	−2.45 ± 0.12 ^a,A^	10.49 ± 0.49 ^a,A^	0.45
	90	1.51 ± 0.04 ^a,B^	75.7 ± 0.8 ^a,A^	31.04 ± 1.09 ^b,A^	−2.35 ± 0.18 ^a,AB^	11.28 ± 0.92 ^a,B^	1.03
2.0	60	4.11 ± 0.08 ^b,B^	86.7 ± 3.6 ^b,A^	30.17 ± 1.83 ^a,B^	−2.34 ± 0.09 ^a,AB^	11.54 ± 0.72 ^b,B^	0.67
	70	3.69 ± 0.07 ^b,A^	90.4 ± 1.4 ^b,A^	29.26 ± 0.73 ^a,AB^	−2.28 ± 0.09 ^ab,BC^	10.97 ± 0.47 ^b,A^	0.84
	80	7.22 ± 0.14 ^b,C^	95.4 ± 1.6 ^b,B^	29.27 ± 0.74 ^a,AB^	−2.41 ± 0.16 ^a,A^	11.54 ± 0.21 ^c,B^	1.05
	90	4.22 ± 0.08 ^b,B^	91.2 ± 2.6 ^b,A^	28.78 ± 1.07 ^a,A^	−2.23 ± 0.04 ^b,C^	11.47 ± 0.20 ^a,B^	1.44
2.5	60	6.80 ± 0.75 ^c,A^	94.4 ± 1.4 ^c,A^	29.55 ± 1.15 ^a,A^	−2.35 ± 0.09 ^a,A^	10.63 ± 0.38 ^a,A^	0.60
	70	10.7 ± 1.28 ^c,B^	99.4 ± 0.5 ^c,B^	29.31 ± 0.81 ^a,A^	−2.37 ± 0.10 ^a,A^	11.92 ± 0.49 ^c,B^	1.30
	80	10.2 ± 1.10 ^c,B^	96.0 ± 1.2 ^b,A^	30.17 ± 1.22 ^b,A^	−2.32 ± 0.10 ^a,A^	11.05 ± 0.56 ^b,A^	0.20
	90	8.78 ± 0.97 ^c,B^	95.5 ± 2.1 ^c,A^	30.39 ± 1.89 ^b,A^	−2.21 ± 0.09 ^b,B^	10.90 ± 0.69 ^a,A^	0.35

^1^ Lowercase letters (a, b, c) indicate statistically significant differences (*p* < 0.05) between gelator concentrations at each drying temperature. Capital letters (A, B, C) indicate statistically significant differences (*p* < 0.05) between the drying temperatures at each gelator concentration. In both cases, Duncan’s test, with comparisons made independently, were performed for each variable.

## Data Availability

Data will be made available on request.

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
