# Peer review of "Characterization of Rapeseed Oil Oleogels Produced by the Emulsion Template Method Using Hydroxypropyl Methylcellulose and the Drying Kinetics of the Emulsions"

_foods, 2025, doi:10.3390/foods14162908_

Round 1
Reviewer 1 Report
Comments and Suggestions for Authors
Review of Characterization of Rapeseed Oil Oleogels Produced by Emulsion Template Method using Hydroxypropyl Methylcellulose and Drying Kinetics of the Emulsions by Mario Lama, Amaya Franco-Uría and Ramón Moreira
This paper investigates the production of oleogels from rapeseed oil using an emulsion-templated approach with hydroxypropyl methylcellulose (HPMC) as the structuring agent, examining how drying temperature and polymer concentration affect their rheological, structural, and functional properties.
Several issues need clarifications:
- The drying step assumes the equilibrium moisture content zero “due to high temperature and low relative humidity,” which is rarely true. This could bias diffusion coefficient calculations. A reviewer may ask for justification or residual moisture measurements.
- Rheological tests use different strain amplitudes for emulsions (1%) and oleogels (0.01%)—the justification for this change is missing. This could affect comparability of G′ and G″ data between stages.
- Textural analysis only uses one compression cycle for hardness, adhesiveness, cohesiveness, elasticity; no mention is made of pre-conditioning or temperature control during testing which can influence results.
- Oxidation tests are performed in duplicate which is low for analytical reliability, especially for peroxide values where variability can be high.
- The colour measurement section uses ΔE relative to a “reference sample” but does not specify what the reference is. Is it fresh emulsion or unprocessed oil?
- Many details that influence results such as exact air humidity control in dryer, actual temperature at sample surface, possible evaporation of volatiles during drying, water content at the end seem rather assumed than measured. Please clarify.
- For OBC tests, the post-centrifugation oil removal method (“inverting for 1 min”) could introduce variability. No mention was made of whether excess oil was blotted or if inversion time was optimized.
- At 60 °C, data was not obtainable, but no explanation for why replications failed is provided. Was it due to instability, microbial growth, or measurement limitations? Please clarify.
- The activation energy for water diffusion in these oleogels could be calculated and discussion on how temperature-sensitive their drying process really is could be included.
- The description of the diffusional model fitting is correct, however it would be useful if some characteristic parameters (R², RMSE) would be reported. Also, please comment on model validity beyond the experimental conditions.
- The statement "no clear trend was observed" for diffusivity vs HPMC concentration could be reframed with more insight. Maybe structural compaction saturates beyond 2% HPMC, limiting further resistance changes, maybe this could be connected to rheology results.
- The strain and frequency sweeps are well interpreted indicating that oleogels are more elastic than emulsions. However, the discussion of LVR shortening could benefit from quantitative detail (for example, compare exact LVR values for 1.5% vs. 2.5% HPMC).
- The observation that moduli increase within temperature interval 60 → 80 °C but drop at 90 °C is interesting, but the syneresis hypothesis is speculative without direct measurement. Providing microstructural evidence would support the explanation.
- The inverse relationship between hardness and adhesiveness is corectly stated, but attributing increased adhesiveness to oil release is plausible, yet not verified here.
- Cohesiveness and elasticity sections seem more as literature surveys than direct interpretation of your dataset. The high variability in elasticity could be quantified with CV% to explain the lack of trends.
- OBC is correlated with modulus and hardness which is a strength of the analysis. However, please list the specific outliers and the corresponding conditions.
- The decline at 90 °C is explained with the same syneresis hypothesis as in rheology section, but again, this is not directly measured, so it should be formulated as a “possible explanation” rather than a conclusion.
- The oxidation section appropriately considers both primary (PI) and secondary (p-anisidine) oxidation indices.The finding of no significant differences is important, but the discussion of drying time vs. temperature effects could be improved by quantifying the drying durations for each temperature.This would make the opposed interaction claim stronger.
- The range of PI values (4.7–7.7) is given without contextual thresholds. Food safety limits should be specified.
- Formatting and style should be revised, English corrections are necessary.
Comments on the Quality of English Language
Some English corrections are necessary.
Reviewer 2 Report
Comments and Suggestions for Authors
- Emulsion drying is a challenging operation where drying methods and operation conditions can give different performance. The process also gives oxidation effect. The advantage should be evaluated with direct methods. More discussion should be provided for the rasoning of the incentives of the study.
- Due to the above reason, a comparison with a direct method product could be interesting for the justification of the discussion.
- Industrial feasibility for the operation could be enhanced in terms of emulsion drying operation to support the incentive of the study.
- The length is relatively too long for such a MS. Reference list could be shortened also. It is a bit redundant for some parts.
- Abstract: For the use of abbreviations, it should be explained to make the reading easier.
Reviewer 3 Report
Comments and Suggestions for Authors
The article Characterization of Rapeseed Oil Oleogels Produced by Emulsion Template Method using Hydroxypropyl Methylcellulose and Drying Kinetics of the Emulsions is well prepared and interesting, but some parts need correction.
Remarks
-How was the low humidity controlled during drying?
-I suggested adding drying rate curves. That is, changes in water content and drying rate.
How was the end of the first drying period determined? I would also provide the drying time for the first and second periods.
Table 2 -Please add the statistical analysis to the colour parameters
-Figure 4 is not clear and it needs improvement (maybe smaller symbols, different colours)
-In the case of hardness, the maximum force in the deformation test is most often used, as described in the authors' methodology, but the force is expressed in N. I can suspect that the authors used the maximum stress (kPa), not maximum force, but it is not the same. It would be necessary to define how this parameter was calculated.
-What was the final dry matter content of the dried samples? This could determine the textural parameters
-The conclusions contain many statements rather than conclusions. In my opinion, there was no final conclusion regarding what the optimal composition and drying conditions were.
Round 2
Reviewer 1 Report
Comments and Suggestions for Authors
Several issues have been clarified.The authors acknowledged few matters that need further investigation, however the manuscript can be published in present form.
Reviewer 3 Report
Comments and Suggestions for Authors The authors addressed all comments. The article has been improved.